# The Role of PIM Kinases in Pediatric Solid Tumors

**DOI:** 10.3390/cancers14153565

**Published:** 2022-07-22

**Authors:** Janet Rae Julson, Raoud Marayati, Elizabeth Ann Beierle, Laura Lee Stafman

**Affiliations:** 1Division of Pediatric Surgery, Department of Surgery, University of Alabama at Birmingham, Birmingham, AL 35233, USA; jjulson@uabmc.edu (J.R.J.); rmarayati@uabmc.edu (R.M.); 2Division of Pediatric Surgery, Department of Surgery, Vanderbilt University, Nashville, TN 37240, USA; laura.stafman@vumc.org

**Keywords:** PIM kinase, pediatric, neuroblastoma, hepatoblastoma, osteosarcoma

## Abstract

**Simple Summary:**

PIM kinases are enzymes that have been identified as potential therapeutic targets in a number of cancers. In this review, we summarize the functions of PIM kinases and the roles they play in the development of cancer, with an emphasis on the discoveries of their functions in pediatric malignancies. We also discuss the currently available drugs targeting PIM kinases and existing clinical trials.

**Abstract:**

PIM kinases have been identified as potential therapeutic targets in several malignancies. Here, we provide an in-depth review of PIM kinases, including their structure, expression, activity, regulation, and role in pediatric carcinogenesis. Also included is a brief summary of the currently available pharmaceutical agents targeting PIM kinases and existing clinical trials.

## 1. Introduction

Provirus integration site for Moloney murine leukemia virus (PIM) kinases have been identified as potential therapeutic targets in several malignancies. Most of what is reported in the literature regarding PIM kinases has been related to adult malignancies or non-cancerous conditions, including hypertension, rheumatoid arthritis, and insulin resistance [1,2,3]. The aim of our work is to provide an in-depth review of PIM kinases, including their structure, expression, activity, regulation, and role in pediatric solid tumor carcinogenesis. We will also discuss the currently available pharmaceutical agents targeting PIM kinases and existing clinical trials.

## 2. PIM Kinase Family

PIM kinases are a family of serine/threonine kinases consisting of three members, PIM1, PIM2, and PIM3, which are encoded by the *PIM1*, *PIM2*, and *PIM3* genes, respectively [4]. PIM genes are located on different chromosomes in the human genome; *PIM1* is located on chromosome 6p21.2, *PIM2* on the X chromosome, and *PIM3* on chromosome 22q13 [4]. Transcripts of *PIM* messenger RNA (mRNA) are encoded by six exons with large 5′ and 3′ untranslated regions (UTRs), containing a GC-rich region and five copies of AUUA destabilizing motifs. Alternative translation initiation sites yield different protein isoforms. Whereas PIM1 and PIM2 exist in two and three isoforms, respectively [5,6], only one isoform of PIM3 has been detected and described [7]. PIM protein isoforms have different molecular masses but retain their serine/threonine kinase activity [8]. There is a high degree of sequence homology between the human PIM proteins at the amino acid level with 71% homology between PIM1 and PIM3, 61% homology between PIM1 and PIM2, and 44% homology between PIM2 and PIM3 [9].

## 3. PIM Kinase Structure

The PIM1 kinase crystal structure has been described as having an N-terminal lobe composed primarily of β-sheets and a C-terminal lobe composed of α-helices with a deep, intervening cleft [10]. The hinge region (residues 121–126) connects the two domains. Several residues in the cleft serve as the ATP binding site, and these residues are conserved between all three PIM kinases. PIM2 has been depicted as having a similarly active conformation on crystallography as PIM1 [11,12], and while the crystal structure of PIM3 kinase has not yet been determined, investigators believe it has a similar structure to that described for PIM1 given the high degree of sequence homology between PIM1 and PIM3.

The PIM kinases are unique from other proteins in that they lack a regulatory domain. Instead, they are constitutively active and regulated on a transcriptional and translational level. Furthermore, the ATP binding pocket in PIM1 is open, regardless of the presence of ATP [10], indicating that PIM1 kinase, as it is expected for PIM2 and PIM3, is maintained in an active state conformation [11,12]. The correlation between elevated protein levels of PIM1 and increased overall kinase activity also provides evidence for the protein’s active state being independent of phosphorylation, as is characteristic of constitutively active enzymes [10]. The stability and function of PIM kinases depend on protein levels achieved by a balance between translation and degradation [13,14].

## 4. PIM Kinases Expression and Regulation

There is evidence that PIM1 expression is developmentally regulated. It is expressed in testis and early embryonic cells, but expression is highest in hematopoietic cells, particularly early in development [15]. PIM1 is highly expressed in the hematopoietic precursors in the fetal liver and spleen but not the corresponding adult tissues [16]. PIM1 knockout mice have a deficiency in interleukin (IL)-7- and steel factor (SLF)-driven pre-B cell growth and IL-3-driven bone marrow-derived mast cell growth [17,18], providing further evidence for the role of PIM1 in hematopoietic lineage development. PIM2 protein is detected highest in brain and lymphocytic cells [19]; *PIM3* mRNA is found in the brain, heart, lungs, kidneys, spleen, and skeletal muscles, as well as in peripheral blood leukocytes [7].

Though tissue expression profiles and physiologic roles for the three members of the PIM family differ, each protein plays a role in growth factor-induced cell survival and proliferation. Triple knockout mice lacking PIM1, PIM2, and PIM3 are viable and fertile; however, they are small and exhibit decreased proliferation of hematopoietic cells in response to growth factors [20]. Hormones, growth factors, and cytokines, including IL-2, IL-3, IL-4, IL-5, IL-6, IL-7, IL-9, IL-12, granulocyte-macrophage colony-stimulating factor (GM-CSF), granulocyte-colony stimulating factor (G-CSF), erythropoietin, interferon-gamma (IFN-γ), SLF, and prolactin [21,22,23,24,25,26,27,28,29], induce the transcription and translation of PIM kinases, mostly via the Janus kinase/signal transducer and the activator of the transcription (JAK/STAT) pathway [30]. A negative feedback loop exists in which PIM1 decreases JAK/STAT signaling by activating the suppressors of cytokine signaling (SOCS) proteins, which in turn inhibit STAT phosphorylation [31]. On the contrary, it has been demonstrated that PIM2 induces a positive feedback loop through the JAK/STAT pathway to increase STAT3 signaling via IL-1α and IL-8 [32].

There are other factors that induce the expression of PIM kinases aside from the JAK/STAT pathway. Nuclear factor kappa-light-chain-enhancer of activated B cells (NFκ-B) is a five-protein complex, which induces the expression of PIM kinases. This protein complex controls DNA transcription, cytokine production, and cell survival in stress-like responses [32,33]. Kruppel-like factor 5 (KLF5) is a transcription factor that induces apoptosis and DNA damage repair and induces the expression of PIM1 [34]. Homeobox A9 (HOXA9), a member of the homeobox family of transcription factors, binds to the *PIM1* promoter and induces the transcription of the gene in hematopoietic cells [35]. Hypoxia also induces the transcription of PIM kinases. In hepatocellular carcinoma cells, hypoxia-inducible factor-1alpha (HIF-1α) and hypoxia-inducible factor-2alpha (HIF-2α) bind directly to the *PIM2* promoter and induce PIM2 expression [36]. PIM2 then interacts with a transactivation domain of HIF-1α, leading to a positive feedback loop between PIM2 and HIF-1α. PIM1 has also been demonstrated to be upregulated in response to hypoxia, although in a HIF-1α-independent manner. In the setting of hypoxia, PIM1 degradation is prevented, and its nuclear translocation is facilitated, leading to higher PIM1 protein levels [37].

PIM1, and likely PIM2 and PIM3, is also regulated at the post-transcriptional level. Cap-dependent translation frequently occurs in GC-rich sequences, which are present in the 5′ UTR of *PIM* mRNAs, and the overexpression of the eukaryotic translation initiation factor 4E (eIF4E) increases PIM1 expression, suggesting that *PIM1* mRNA is translated in a cap-dependent manner [38]. The binding of eIF4E to a stem-loop pair sequence present in the 3′ UTR of *PIM1* mRNA allows for the nuclear export and translation of *PIM1* transcript [39]. Since *PIM3* mRNA shows high sequence homology to *PIM1* mRNA, eIF4E may regulate the translation of *PIM3* mRNA in a similar manner.

The 3′ UTR of *PIM1* and *PIM3* harbors multiple putative target sites for micro RNAs (miRNAs) that may also act to suppress PIM1/3 expression at the post-transcriptional level. Employing a dual luciferase assay screen, Du et al. found several miRNAs decreased PIM3 expression, but only miR-506 functioned as a tumor suppressor and correlated negatively with PIM3 expression in pancreatic cancer tissues [40]. Other investigators demonstrated that miR-33 targets *PIM3* mRNA and that the overexpression of miR-33a in pancreatic cancer cell lines suppressed PIM3 expression [41,42].

Because PIM kinases are constitutively active once expressed, protein stability and degradation significantly contribute to the cellular function of these proteins. Each PIM kinase, and even each isoform, has a different half-life, but in general the half-lives are short, ranging from minutes to one hour [43,44]. Protein phosphatase 2A (PP2A) dephosphorylates PIM1 and PIM3 [45,46], leading to ubiquitination and proteasomal degradation [47,48]. PIM2 is degraded in a similar fashion, though it does not necessarily require dephosphorylation [49]. Heat shock proteins (Hsp) also regulate PIM kinase stability, with Hsp70 promoting PIM1 degradation and Hsp90 protecting PIM1 from proteasomal degradation [50].

## 5. Role of PIM Kinases in Carcinogenesis

PIM1 expression is associated with pancreatic, prostate, gastric, and colorectal cancers, as well as bladder and squamous cell carcinomas and liposarcoma [19,51,52,53]. PIM1 and PIM2 expression are correlated with hematologic malignancies, including myeloma, lymphoma, and leukemias [54,55]. Normal endoderm-derived organs, such as the liver, stomach, small intestine, colon, prostate, or ovary, do not express *PIM3* mRNA [7]. However, PIM3 has been shown to be highly expressed in premalignant lesions and malignant tumors of these organs [7,56]. In the stomach and colon, PIM3 protein was detected with a higher incidence in adenomatous tissues than in adenocarcinoma tissues [57,58]. Similarly, in liver lesions with malignant potential, such as regenerative nodules and adenomatous hyperplasia, PIM3 protein was detected at a higher frequency than in hepatocellular (HCC) cells [7]. These findings suggest that aberrant PIM3 expression may occur as an early event in carcinogenesis.

The major downstream effects of PIM kinase signaling include: (i) increased cancer cell survival and decreased apoptosis through the phosphorylation of pro-apoptotic protein BCL2 associated agonist of cell death (Bad), (ii) increased proliferation via the regulation of cell cycle progression, (iii) the regulation of Myc transcriptional activity, (iv) the promotion of cellular motility and metastasis, and (v) maintaining cancer cell stemness, all of which are consistent with promoting tumorigenesis and tumor maintenance (Figure 1)

### 5.1. Apoptosis

One of the main downstream effects of PIM kinases is the inhibition of apoptosis [55,59,60]. All three PIM kinases phosphorylate Bad [55,59,60], a regulator of apoptosis. When phosphorylated, Bad is sequestered in the cytoplasm and cannot bind to B-cell lymphoma 2 (Bcl-2), thus leaving Bcl-2 free to perform its anti-apoptotic function [55,59]. The knockdown of PIM3, using short interfering RNA (siRNA) or short hairpin RNA (shRNA), has been shown to enhance apoptosis and decrease in vitro survival of various types of human cancer cells, such as colon, pancreatic, and HCC [7,54,60,61]. Conversely, the forced expression of PIM3 kinase in colon and pancreatic cancer cells increases the amount of phosphorylated Bad [54,60], inhibiting apoptosis and promoting carcinogenesis.

### 5.2. Cell Cycle

PIM kinases regulate the cell cycle via multiple mechanisms. The G1 checkpoint cyclin-dependent kinase inhibitor 1B (CDKN1B, p27kip1, or p27) plays a crucial role in tumor suppression by binding to and preventing the activation of cyclin E-CDK2 complexes, thus inhibiting abnormal cell cycle progression [62]. All three PIM kinases bind to and phosphorylate p27 at its threonine residues, which promotes the interaction of p27 with 14-3-3 protein, resulting in its nuclear export and proteasomal degradation [62]. PIM1 also increases degradation of p27 by phosphorylating and stabilizing S-Phase kinase associated protein 2 (Skp2), the E3 ligase that ubiquitinates p27 [63]. In addition to the posttranslational regulation, Morishita et al showed that PIM1 suppressed *p27* gene transcription via the phosphorylation and inactivation of forkhead transcription factors FoxO1a and FoxO3a [62]. By downregulating p27 expression via these mechanisms, PIM kinases promote cell cycle progression at the G1 phase, allowing for increased proliferation.

Another mechanism by which PIM kinases regulate cell cycle progression is through the phosphorylation of CDKN1A (p21waf1, or p21) at its threonine residues. Phosphorylation of p21 leads to its accumulation in the cytoplasm and enhances its stability [64,65,66]. Without p21 in the nucleus, cyclin E-CDK2 and D-CDK4 complexes are unchecked, leading to the progression of the cell cycle at the G1 phase.

PIM kinases also affect cell cycle by phosphorylating members of the cell division cycle 25 (Cdc25) tyrosine phosphatase family, particularly Cdc25A and Cdc25C [67,68,69,70]. By phosphorylating Cdc25A, PIM1 increases the phosphatase activity of Cdc25A and its activation of cyclin D1-associated kinases, resulting in cell cycle progression [67]. PIM kinases also phosphorylate the Cdc25C-associated kinase 1 (C-TAK1), which is a potent inhibitor of Cdc25C [70]. By phosphorylating C-TAK1, PIM1 inhibits C-TAK1′s inhibition of Cdc25C, a protein that actively promotes cell cycle progression at the G2/M phase, thereby advancing the cell cycle [70].

### 5.3. C-Myc Regulation

PIM kinases promote tumorigenesis by regulating the transcriptional activities of the proto-oncogene c-Myc. Investigators have proposed several mechanisms to explain the cooperation between Myc and PIM kinases. Olivero and colleagues suggested that PIM1 phosphorylates Ser10 of histone H3 on the nucleosome at the Myc-binding sites, acting as a co-activator of Myc and contributing to Myc-dependent transcriptional activation [71]. Consistently, expression profile analyses demonstrated that PIM1 contributes to the regulation of 20% of the Myc-regulated genes [71]. Other investigators showed that PIM1 and PIM2 regulate the phosphorylation of c-Myc protein at its serine (Ser329, Ser62) and threonine (Thr58) residues [72], resulting in the stabilization and subsequent enhancement of the transcription activities of c-Myc protein. PIM3, specifically, enhanced *c-Myc* mRNA expression via the activation of PGC-1α [73]. Together, these findings indicate that PIM kinases contribute to the oncogenic transformation of cells by modulating *c-Myc* transcriptional activity.

### 5.4. Motility

Another important role of PIM kinases is the promotion of cellular motility. PIM kinases regulate the surface expression of CXCR4 [74,75,76], which upon interaction with its ligand, C-X-C motif chemokine ligand 12 (CXCL12) promotes migration, invasion, and metastasis [75]. In prostate cancer, PIM1 has been shown to play a role in metastases through a number of pathways, including (i) the activation of NFATC1, c-Myc, and Smad signaling, (ii) the inhibition of GSK3β and FOXP3, (iii) the upregulation of PTGS2, and (iv) altering integrin-dependent cell adhesion [75,77,78,79]. The overexpression of PIM1 in nasopharyngeal carcinoma cells also led to increased migration [80] with similar findings regarding PIM2 in non-tumorous liver cell lines, but mechanisms were not elucidated in either study [81]. Uddin et al. showed that PIM2 played a role in the epithelial–mesenchymal transition of breast cancer, a critical step in promoting cellular motility and ultimately metastases, which they hypothesized to occur through the persistent activation of STAT3, driven by PIM2 activation of IL-1α and IL-8 [32]. PIM3 is highly expressed in endothelial cells and plays a role in endothelial cell spreading, migration, and the formation of tube-like structure necessary for angiogenesis. PIM3 co-localizes with the adhesion protein focal adhesion kinase (FAK) in lamellipodia, which are cellular protrusions that format the leading edge of spreading endothelial cells, indicating a role in motility and migration [82].

### 5.5. Cancer Cell Stemness

Stem cell-like cancer cells (SCLCCs) are a subpopulation of cancer cells with characteristics of pluripotent stem cells thought to be responsible for tumor recurrence and chemotherapeutic resistance. Although the concept of SCLCCs is well studied, only a handful of researchers have investigated the role and importance of PIM kinases in maintaining this subpopulation of cells through effects on gene expression, cell signaling pathways, and the tumor microenvironment (TME).

PIM kinases have been shown to correlate with cancer stemness genes. Jiménez-García et al reported PIM1 and PIM2 protein expression correlated with the expression of genes related to cancer stem cells and pluripotency in breast, ovarian, and prostate cancer [83,84]. Others have shown that *PIM1* and *PIM2* mRNA expression correlated with mRNA expression of *CD44* and *CD133*, two markers of glioblastoma cancer stem cells [85,86]. Studies in prostate cancer found PIM1 affected stem cell proliferation, self-renewal, and expansion [87]. Li et al demonstrated PIM3 is overexpressed in pancreatic SCLCCs and its expression correlated with stemness-associated surface markers [88].

The correlation between PIM kinases and a stem cell signature may contribute to the therapeutic resistance found in tumors overexpressing PIM kinases. The inhibition of PIM3 kinase increased the sensitivity of pancreatic cancer cells to both chemotherapy and radiotherapy [41,89,90]. Similar findings regarding chemotherapy sensitivity have been reported with PIM1 kinase inhibition in prostate cancer [91,92] and PIM2 kinase inhibition in ovarian cancer [93]. Guo et al reported that PIM3 was one of three most highly expressed proteins in adriamycin- and vincristine-resistant gastric cancer cell lines, and silencing of PIM3 reversed the adriamycin-resistant phenotype [94]. In HCC, treatment with the chemotherapeutic agents doxorubicin, 5-fluorouracil, or cisplatin, induced PIM3 expression in patient tumors [95]. Additionally, PIM3 overexpression generated multidrug resistance (MDR) in vitro, and PIM3 suppression with Ubenimex reversed MDR and the enhanced cisplatin-induced apoptosis of HCC cells [95].

The mechanism by which PIM3 kinase influences stemness and drug resistance is thought to be through effects on self-renewal and reprogramming pathways, including STAT3, Nanog, Oct4, c-Myc, and β-catenin signaling. PIM3 overexpression induced stemness in pancreatic cancer cells via the activation of the STAT3 pathway [88]. The overexpression of PIM3 in prostate cancer cells led to increases in *NANOG* and *OCT4* gene expression [58]. Together with Sox2, Nanog and Oct4 form an intricate regulatory loop to activate the transcription of genes that support pluripotency. PIM kinases consistently phosphorylate Oct4 and Myc, which also contributes to the nuclear reprogramming of pluripotent cancer stem cells [87,96,97]. Liang et al showed that PIM3 promoted gemcitabine resistance by activating AKT/β-catenin signaling [41]. The *MDR1* gene, a downstream target of β-catenin [98], encodes the P-glycoprotein transporter responsible for the active efflux-pumping of antineoplastic agents. Thus, it is hypothesized that PIM3 may contribute to *MDR1* expression and gemcitabine resistance in HCC cells by up regulating β-catenin [95].

Alternative to the regulation of transcription factors in cancer cells, PIM kinases may also influence self-renewal and reprogramming by inducing a pro-inflammatory TME [84]. Monocytes and macrophages recruited to tumors directly regulate SCLCCs through the inflammatory cytokines IL-1, IL-6, and IL-8, which drive self-renewal in SCLCC niches. These cytokines produced by the TME activate the STAT3/NF-κB pathways in tumor and stromal cells, generating positive feedback loops that contribute to SCLCC self-renewal [99].

## 6. Role of PIM Kinases in Pediatric Malignancies

In pediatric malignancies, the role of PIM kinases has been most extensively studied in hepatoblastoma and to a lesser extent in neuroblastoma and osteosarcoma. PIM3 is the specific family member believed to play a role in hepatoblastoma tumorigenesis and tumor maintenance. PIM1, on the other hand, is the PIM kinase most extensively studied in neuroblastoma and osteosarcoma.

### 6.1. Hepatoblastoma

Stafman et al were the first to examine the role of PIM3 kinase in hepatoblastoma and to propose PIM inhibition as a potential therapeutic approach to this malignancy. They demonstrated that PIM3 kinase is expressed in both the human hepatoblastoma established cell line, HuH6, and in a human hepatoblastoma patient-derived xenograft (PDX) [100]. They also assessed the expression of PIM3 kinase in 19 human hepatoblastoma patient specimens using immunohistochemistry and found that 74% of samples expressed PIM3, whereas normal liver tissue did not [100,101]. In that study, higher PIM3 expression correlated with significantly worse patient survival independent of histologic subtype, alpha-fetoprotein levels, or tumor stage [101].

The knockdown of PIM3 with siRNA or treatment with the pan-PIM inhibitors AZD1208 or PIM447 resulted in decreased hepatoblastoma cell viability, motility, and attachment-independent growth [100,102]. Further, PIM kinase inhibition with AZD1208 led to decreased phosphorylation of p21 at the Thr145 site, resulting in cell cycle arrest. AZD1208 also induced apoptosis as evidenced by the increase in the cleavage of PARP and caspase 3 and the decreased phosphorylation of Bad at the Ser112 site [100].

In a subcutaneous xenograft model of hepatoblastoma, mice treated with AZD1208 at 30 mg/kg had significantly smaller tumors than those treated with a vehicle control [100]. Further, PIM3-overexpressing HuH6 cells yielded significantly larger tumors compared to empty vector control cells in a murine flank model [101]. Marayati et al. further explored the role of PIM3 kinase in hepatoblastoma tumorigenesis, establishing a PIM3 knockout cell line using CRISPR-Cas9 technology [103]. Similar to the previously described findings with siRNA or pan-PIM inhibition, they showed that PIM3 knockout cells had significantly decreased proliferation and motility and were arrested in G0/G1 phase of the cell cycle. Mice injected with PIM3 knockout cells had decreased tumor volumes and their tumors demonstrated decreased Ki67 staining, denoting decreased proliferation, compared to animals bearing tumors from wild-type cells [103].

Stafman et al performed seminal studies examining the role of PIM3 kinase in hepatoblastoma cancer cell stemness [101]. They showed that PIM3-overexpressing hepatoblastoma cells formed tumorspheres more readily than empty vector control cells, which is a characteristic of SCLCCs [101]. They corroborated these findings when they demonstrated decreased tumorsphere formation and CD133 cell surface expression, a marker of stemness, in PIM3 knockout cells [100]. The mRNA abundance of stemness markers *Oct4*, *Nanog*, *Sox2*, and *Nestin* was decreased in the PIM3 knockout cells [103]. Small molecule PIM inhibitors AZD1208 and PIM447 also inhibited the SCLCC phenotype in hepatoblastoma cells [100,102]. In vivo*,* the pharmacologic inhibition of PIM kinases decreased tumor growth in mice implanted with hepatoblastoma SCLCCs, with 57% of the tumors completely regressing, providing further evidence that PIM inhibition decreases hepatoblastoma stemness [100].

Studies have documented that cisplatin treatment selects for a SCLCC subpopulation of cancer cells that are chemo-resistant [103]. There is evidence that PIM kinases play a role in cisplatin resistance in hepatoblastoma. Marayati and colleagues demonstrated that the stable overexpression of PIM3 in hepatoblastoma cell lines led to increased resistance to cisplatin [102,104]. To further explore the role of PIM3 in chemoresistance, Marayati et al developed cisplatin-resistant hepatoblastoma cells using a xenograft model [104]. These cisplatin-resistant hepatoblastoma cells had increased PIM3 kinase expression and SCLCCs properties. When treated with AZD1208, the resistant cells were re-sensitized to cisplatin [104]. Combining PIM inhibitors with cisplatin was synergistic in vitro and decreased hepatoblastoma tumor growth in vivo more effectively than either drug alone [100,102], suggesting that PIM inhibition may be employed as a novel therapeutic adjunct to eradicate the SCLCC that may be contributing to drug resistance.

### 6.2. Neuroblastoma

In 2018, Brunen et al. first described PIM kinases as a potential prognostic biomarker and therapeutic target for neuroblastoma [105]. In their review of the Kocak dataset (Gene Expression Omnibus GSE45547, *n* = 476), they found high expression of either *PIM1*, *PIM2*, or *PIM3* mRNA to be significantly associated with poor overall survival in neuroblastoma. Additionally, they demonstrated that *PIM1* and *PIM2* expression were associated with more advanced disease. The siRNA knockdown of *PIM1* reduced the viability of two neuroblastoma cell lines, KELLY and SH-SY5Y. They described PIM inhibition-resistant neuroblastoma cell lines, based on the high median lethal dose (LD_50_), and utilized a genome-wide CRISPR-Cas9 genetic screen to explore the mechanism of PIM inhibitor resistance. They identified and confirmed the loss of *NF1* to be associated with resistance to PIM inhibition. In an in vivo model, they showed that the AZD1208 treatment of mice injected with *NF1* knockout KELLY cells had decreased survival, indicating decreased sensitivity to AZD1208, compared to those with *NF1* wild-type tumors.

Similar to Brunen et al., Trigg et al. found some neuroblastoma cell lines to be PIM inhibition-resistant, leading the investigators to propose that PIM kinase inhibition alone may not be a viable treatment option in neuroblastoma, but could be employed in combination with anaplastic lymphoma kinase (ALK) inhibition [106]. Through CRISPR-Cas9 screening, they identified PIM1 as a clinically relevant target that may promote resistance to ALK inhibitors. They demonstrated that PIM1-mediated ALK inhibitor resistance occurred through the phosphorylation of BAD, resulting in decreased neuroblastoma apoptosis. Additionally, they showed that PIM kinase inhibition using AZD1208 acted in a synergistic manner with ALK inhibitors in both established cell lines in vitro, as well as in an in vivo PDX model [106].

Other researchers have provided evidence that PIM kinase inhibition will be more effective in neuroblastoma if administered in combination with other therapeutic agents. Mohlin et al. investigated IBL-301 and IBL-302, novel multi-kinase inhibitors, directed towards PIM1-3, PI3K, and mTOR [107]. They found that treatment with IBL-301 increased the differentiation of neuroblastoma cells as demonstrated by neurite outgrowth and an increase in *GAP43*. They also demonstrated increased cell death via apoptosis in neuroblastoma PDX cells treated with IBL-302. These investigators evaluated combination therapies in vitro and found that IBL-302 enhanced the effects of common neuroblastoma therapeutics, including etoposide, cisplatin, and doxorubicin. They studied combination therapy in vivo comparing IBL-302 alone, cisplatin alone, or a combination of both drugs, and found that animals treated with the combination had decreased or no tumor growth and a statistically significant increase in survival. Additionally, MYCN, a marker of high-risk disease, was decreased in the tumors evaluated by immunohistochemistry after treatment with combination therapy.

### 6.3. Osteosarcoma

Liao et al. found elevated PIM1 protein expression in patients with metastatic osteosarcoma at the time of initial presentation, and these patients had significantly lower 5-year survival (34% vs. 84.2%, *p* < 0.001) [108]. The investigators ultimately showed that PIM1 expression independently predicted overall survival and disease-free survival in osteosarcoma [108]. Narlik-Grassow similarly found that PIM1, but not PIM2 or PIM3, expression correlated with poorer prognosis in osteosarcoma [109]. PIM1 inhibition in osteosarcoma cells by siRNA [108,109] or the PIM1 inhibitor, SMI-4a [108], led to increased apoptosis, and decreased proliferation and motility. Overexpressing PIM1 in the human osteosarcoma cell line U2-OS led to increased proliferation and motility, implying a more metastatic phenotype [109]. The researchers hypothesized that PIM-induced alterations in cyclin D1 were responsible for changes in proliferation and that PIM effects on matrix-metalloprotease 2 (MMP2) affected motility [109].

Liu et al. studied the role of PIM1 kinase in osteosarcoma in relations to the tumor suppressor, miR-486 [110]. Their database review demonstrated a negative correlation between PIM1 and miR-486 (r = −0.5445, *p* < 0.001). Similar to Narlik-Grassow et al., they performed siRNA knockdown of PIM1 in the MG63 osteosarcoma cell line and found decreased invasion, which was rescued with the re-expression of PIM1. They found that miR-468 binds to *PIM1* mRNA 3′-UTR and hypothesized that this interaction is how miR-468 potentially mediates its action on PIM1.

Narlik-Grassow further explored the impact of the other two PIM kinase family members in osteosarcoma using mouse embryonic fibroblasts (MEFs) generated from PIM2/3 knockout and PIM triple knockout (TKO) mice [109]. They chemically induced tumors in wild-type, PIM2/3 knockout, and TKO mice. TKO mice required a longer time to develop tumors compared to PIM2/3 knockout or wild-type mice (32, 21, and 9 days, respectively). Microscopic examination showed more bone invasion in tumors from the wild-type (80%), compared to the PIM2/3 knockout (31%) or TKO (17%) mice. They also provided support for the effects of PIM on cell cycle progression, as shown by a decrease in cyclin D1 in tumors from PIM TKO mice. Additionally, they demonstrated Gsk3β phosphorylation to be eliminated in tumors from PIM2/3 knockout and TKO mice. The phosphorylation of Gsk3β inactivates the enzyme, resulting in cell cycle progression. Thus, the increased activity of Gsk3β in TKO mice might explain the delayed time to tumor development and further implies PIM kinases play a role in the Gsk3β pathway.

## 7. Pharmacologic Inhibition of PIM Kinases

### 7.1. Therapeutic Agents

There are no inhibitors specific for a particular member of the PIM kinase family currently in clinical trials, but a few experimental compounds have been described [111,112,113]. Researchers have primarily focused on pan-PIM inhibitors, which inhibit all three family members (PIM1, PIM2, and PIM3). Most of these compounds are competitive inhibitors targeting the ATP-binding pocket to inhibit protein function. Many pan-PIM kinase inhibitors have been described, including SGI-1776, TP-3654 (SGI-9481), CX-4945, LY-335′531 (Ruboxistarin), CXR-1002, LY-2835219 (Abemaciclib), INCB053914 [113], CX-6258 [114], LGB321 [115] GDC-0339 [116], GNE-955 [117], AZD1208 [118], PIM447 (LGH447) [119], and MEN1703/SEL24 [120]. Kaewchim et al created a bacteria-derived recombinant PIM2 (rPIM2), which they described as being equally effective at inhibiting PIM2 kinase as AZD1208, though the studies have not yet been advanced to in vivo models [121]. PIM inhibitors along with their respective clinical trials are summarized in Table 1.

### 7.2. Clinical Trials

SGI-1776 was the first PIM kinase inhibitor employed in a human clinical trial, being studied for the treatment of castration-resistant prostate cancer and relapsed/refractory non-Hodgkin lymphoma in adults [113]. The trial was terminated due to QTc prolongation in some patients. This finding was due to the metabolites of SGI-1776 and not a result of PIM inhibition, thus suggesting promise for future trials of PIM kinase inhibitors [13].

AZD1208 has been evaluated in two phase I clinical trials. A phase I clinical trial (NCT01588548), completed in 2014 in Japan and the United Kingdom, assessed the safety and tolerability of increasing doses of AZD1208 in adult patients with advanced solid malignancies and malignant lymphoma [122]. There were no serious adverse events at the highest dose of 800 mg daily. The AZD1208 studies did not detect the QTc prolongation previously observed with SGI-1776. A second phase I clinical trial (NCT01489722) began in 2012 in the United States and Canada to assess the safety, tolerability, pharmacokinetics, and efficacy of AZD1208 in adult patients with recurrent or refractory acute myelogenous leukemia (AML). Investigators terminated this trial in 2014 and no results have been posted.

PIM447 is a more potent pan-PIM kinase inhibitor compared to AZD1208 [123] and has recently been evaluated in preclinical studies in myeloma and other lymphoid tumors [124]. Results from a phase I clinical trial (NCT02160951) in adult patients with relapsed or refractory multiple myeloma demonstrated single-agent anti-tumor response and a tolerable safety profile when administered at doses up to 300 mg daily [125]. A subsequent dose-escalation study in this population was terminated early, citing sponsor decision to focus on other combination therapies and indications for PIM447. This was despite the drug being well tolerated, as the overall risk reduction ratio was not significant enough to support the use of PIM447 as a monotherapy in this population [125].

Two other pan-PIM inhibitors have been evaluated in clinical trials. INCB053914 was evaluated in relapsed and refractory diffuse large B cell lymphoma, while a similar study in relapsed and refractory multiple myeloma was withdrawn due to lack of funding [114]. A study in advanced stage solid tumors was terminated, citing “strategic business decisions” despite the drug being well-tolerated with minimal side effects [126]. CXR-1002 has also been advanced to phase I trials and is well tolerated, even at doses up to 750 mg/kg weekly [114]. No results have been posted for these studies.

### 7.3. Combination Therapies including PIM Kinase Inhibitors

There are currently drugs undergoing investigation in clinical trials that have either been designed for the inhibition of PIM kinases in combination with other targets or were subsequently found to have such properties [113]. One example is CXR-4945, a dual PIM1 and casein kinase-2 (CK2) inhibitor, which has undergone Phase I trials in adults with multiple myeloma, breast cancer, and solid tumors. Another example is LY-2835219, which inhibits both PIM1 and CDK4/6 and is unique in its ability to cross the blood–brain barrier [113]. A total of 81 clinical trials have been conducted with LY-2835219, including 39 phase I trials, 31 phase II trials, 11 phase III trials, in liposarcoma, breast, and prostate cancer, and 3 phase IV trials in breast cancer [125]. Of these, there are three phase I trials in pediatric patients currently underway for the treatment of metastatic or unresectable nuclear protein of the testis (NUT) midline carcinoma, diffuse intrinsic pontine glioma, and several recurrent or refractory solid tumors, including brain, neuroblastoma, Ewing sarcoma, rhabdomyosarcoma, osteosarcoma, and rhabdoid tumors [125]. SEL24/MEN1703 is a third dual PIM kinase inhibitor, which functions as a combination pan-PIM inhibitor and FLT3 inhibitor [120]. It is currently being studied in phase I/II trials (NCT03008187) in patients with acute myeloid leukemia. Initial study data suggest it has an acceptable safety profile and is particularly efficacious in patients with isocitrate dehydrogenase gene (*IDH*) mutations [126,127,128].

## 8. Conclusions

The PIM kinase family plays numerous roles in carcinogenesis. In this review, we outline these functions, including their roles in apoptosis, cell cycle progression, regulation of c-Myc, motility, cancer cell stemness, and chemoresistance, and include summaries of the current understanding of PIM kinases in pediatric solid tumors. Several PIM kinase inhibitors and multi-kinase inhibitors have been shown to be safe in clinical trials and warrant further evaluation in children. As researchers strive to develop new therapeutic agents for the treatment of pediatric solid tumors, particularly for children with resistant or relapsed disease, PIM kinases remain a promising target.

## Figures and Tables

**Figure 1 cancers-14-03565-f001:**
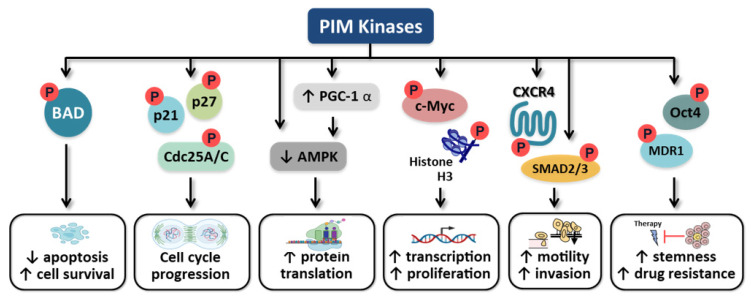
Presumed roles of PIM kinase in carcinogenesis. Targets of PIM kinase phosphorylation and their functional outcome. BAD: BCL2 associated agonist of cell death; AMPK: AMP-dependent protein kinase; PGC-1α: peroxisome proliferator-activated receptor-gamma coactivator 1α.; CXCR4: C-X-C chemokine receptor type 4; SMAD2/3: SMAD family member 2/3; MDR1: multi-drug resistance-1. Created with BioRender.com (accessed on 14 July 2022).

**Table 1 cancers-14-03565-t001:** Current pharmacologic agents for PIM kinase inhibition in clinical trials organized by drug type and malignancy. The phases of clinical trials in which the drug has been tested are summarized and National Clinical Trial (NCT) numbers are included (www.clinicaltrials.gov) (accessed on 14 July 2022).

Pharmacologic Agent	Cancer Type or Condition	Clinical Trial Phases Conducted	Clinical Trial Number(s)
AZD1208	Acute myelogenous leukemia	Phase I (Terminated)	NCT01489722
	Advanced solid tumors, malignant lymphoma	Phase I (Completed)	NCT01588548
PIM447	Myelofibrosis	Phase I (Completed)	NCT02370706
	Multiple myeloma	Phase I (Completed)	NCT02160951
	Relapsed and/or refractory multiple myeloma	Phase I (Completed)	NCT02144038
	Acute myeloid leukemia, high risk myelodysplastic syndrome	Phase I (Completed)	NCT02078609
	Relapsed and/or refractory multiple myeloma	Phase I (Completed)	NCT01456689
SGI-1776	Relapsed and/or refractory leukemias	Phase I (Withdrawn)	NCT01239108
	Prostate cancer, Non-Hodgkin lymphoma	Phase I (Terminated)	NCT00848601
INCB053914	Relapsed and/or refractory multiple myeloma	Phase I (Withdrawn)	NCT04355039
	Advanced solid tumors	Phase I/II (Terminated)	NCT02587598
	Relapsed and/or refractory diffuse large B cell lymphoma	Phase I (Completed)	NCT03688152
CX-4945	Cholangiocarcinoma	Phase I/II (Completed)	NCT02128282
	Multiple myeloma	Phase I (Unknown)	NCT01199718
	Advanced solid tumors, breast cancer, inflammatory breast cancer, Castleman disease, multiple myeloma	Phase I (Unknown)	NCT00891280
SEL24/MEN1703	Acute myeloid leukemia	Phase I/II (Recruiting)	NCT03008187
LY-2835219 (Abemaciclib)	Relapsed or refractory pediatric solid tumors	Phase I (Recruiting)	NCT04238819, NCT02644460
	Pediatric brain tumors	Phase I (Recruiting)	NCT02644460
	Advanced malignancies	Phase I (Active, Completed)	NCT02117648, NCT01394016, NCT02919696, NCT04071262, NCT02857270, NCT01655225, NCT05307705, NCT02791334, NCT02745769
	Breast cancer, metastatic breast cancer	Phase I (Active, Completed), Phase II (Active, Completed, Recruiting), Phase III (Active, Completed, Recruiting), Phase IV (Recruiting, Terminated, Withdrawn)	NCT02831530, NCT02441946, NCT02246621, NCT02102490, NCT05169567, NCT02763566, NCT04752332, NCT03988114, NCT02057133, NCT03703466, NCT03763604, NCT02779751, NCT02675231, NCT02792725, NCT03155997, NCT02747004, NCT04031885, NCT02107703, NCT03130439, NCT04707196, NCT05169567, NCT04975308, NCT04188548, NCT02784795, NCT02688088, NCT04305834, NCT03979508, NCT05307705, NCT04481113, NCT04351230, NCT04256941, NCT03878524
	Non-small cell lung cancer, metastatic non-small cell lung cancer	Phase I (Active, Completed), Phase II (Completed), Phase III (Active)	NCT02079636, NCT02450539, NCT02779751, NCT02152631, NCT02411591
	Sarcoma, dedifferentiated liposarcoma	Phase II (Active), Phase III (Recruiting)	NCT02846987, NCT04967521
	Brain tumor, recurrent glioblastoma	Phase II (Active, Withdrawn)	NCT03220646, NCT02981940, NCT04118036
	Metastatic castration-resistant prostate cancer	Phase II (Active, Recruiting), Phase III (Recruiting)	NCT04408924, NCT03706365, NCT05288166
	Mantle cell lymphoma	Phase II (Active)	NCT01739309
	Pancreatic ductal adenocarcinoma	Phase II (Completed)	NCT02981342
	Metastatic breast or non-small cell lung cancer, or melanoma with brain metastasis	Phase II (Completed, Withdrawn)	NCT02308020, NCT04585724
	Small cell lung cancer	Phase I (Recruiting)	NCT04010357
	Metastatic cancer, *BRAF V600E*, *MEK1*, *MEK2*, *ERK*, *KRAS*, or *RAF1* gene mutations	Phase II (Recruiting)	NCT04534283
	Endometrial cancer, metastatic endometrial cancer	Phase I (Recruiting), Phase II (Active)	NCT04188548, NCT04049227, NCT04469764
	Metastatic or locally advanced anaplastic/undifferentiated thyroid cancer	Phase II (Recruiting)	NCT04552769
	Unresectable of metastatic colorectal cancer	Phase I/II (Recruiting)	NCT04616183
	Mesothelioma	Phase II (Recruiting)	NCT03654833
	Multiple myeloma	Phase I/II (Recruiting)	NCT03732703

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
