# Peer review of "The Role of PIM Kinases in Pediatric Solid Tumors"

_cancers, 2022, doi:10.3390/cancers14153565_

Round 1

Reviewer 1 Report

In this manuscript by Julson JR and Marayati R the role of PIM kinases in pediatric malignancies in reviewed.

The manuscript is concise and well-written and provides a good overview of the subject. The structure is well-organized and language is appropriate.

I have several comments which I would like to ask the authors to address:

-          Line 130: I have a different experience with PIM3 from the one reported here. Our IHC stainings have clearly indicated some expression of PIM3 In normal liver and colon (unpublished observations). I wonder if there are some gene expression datasets that would confirm the alleged lack of mRNA for PIM3 in normal endoderm-derived organs. It is a rather ubiquitous kinase.

-          There is a mistake in citation number 68 – this paper has been authored by Alessio Zippo, Oliviero S. was the corresponding author

-          The pharmacologic inhibition section needs to be updated – there is another clinically tested PIM inhibitor, MEN1703/SEL24 which is moving into phase 2 clinical trial in AML.

-          In addition, I would like to recommend also mentioning the role of PIM kinases in B cell malignancies and synergy with immunotherapy. E.g. PIM kinase targeting has been shown to be effective in models of Burkitt lymphoma and DLBCL, where it synergized with rituximab (10.1158/0008-5472.CAN-21-1023). Since Burkitt lymphoma is a pediatric malignancy, this would warrant a mention in this review.

Reviewer 2 Report

This is a well written review on PIM and I would accept it as is with one caveat.

I would encourage the authors with some little more effort to expand the review to all oncological and non-oncological indications where PIM inhibitors have been used to date including of course the pediatric malignancies ?

It would be a pity not to do that as the rest of the PIM review is quite extensive and well laid out.

One side of the coin is that if the authors want to describe the roles of PIM kinases in pediatric malignancies why do the authors describe in all details the involvement of PIM in carcinogenesis and cancer in general ? All this could be abbreviated by referring to published reviews. Also why adding all PIM inhibitors with the corresponding cancer trials (with 2-3 trials addressing pediatric malignancies). Focusing the review on PIM kinases in pediatric malignancies would shorten the review quite substantially but it would at least fit the title.

The other side of the coin is that the authors have done a thorough review of the rationale to inhibit PIM kinases not only in pediatric malignancies but also in other cancers as well. This would ask for 2-3 additional chapters covering for example the role of PIM inhibition in solid tumors and hematological malignancies. All this would better fit to the introductions and careful review of the PIM kinases in malignancies in general.

Please consider my review more as a suggestion rather than a must.
In any case as the manuscript stands now it is not focusing on the role of PIM kinases in pediatric malignancies.

Reviewer 3 Report

Authors present the role of PIM kinases in childhood cancers. The paper is interesting and I accept after minor corrections.

In the Introduction section: The first sentence: The abbreviation "PIM kinase" should be clarified due to the fact that the authors are using it for the first time.

I suggest extending the Introduction section.

On lines 236-237 – please, remove the bold.

References:

number 100: please, remove the bold;

number 124: this is not complete;

number 126: this is not complete, the year should be bold
